# More Worker Capped Brood and Honey Bees with Less Varroa Load Are Simple Precursors of Colony Productivity at Beekeepers’ Disposal: An Extensive Longitudinal Survey

**DOI:** 10.3390/insects13050472

**Published:** 2022-05-18

**Authors:** Andre Kretzschmar, Alban Maisonnasse

**Affiliations:** 1INRA, BioSP-Biostatistics and Spatial Processes, 228 Route de l’Aérodrome, CEDEX 9, 84914 Avignon, France; 2ADAPI, Maison des Agriculteurs, 22 Avenue Henri Pontier, CEDEX 1, 13626 Aix-en-Provence, France; a.maisonnasse.adapi@free.fr

**Keywords:** honey bee, productivity, colonial structure, Varroa load, lavender honey flow, France

## Abstract

**Simple Summary:**

Annual and regional variations in lavender honey flow are significant in the southeast of France. Beekeepers are wondering how they can buffer these variations by modifying the populations’ parameters of colonies. For 13 consecutive years (2009–2021), colony population parameters and Varroa load were measured to identify the importance of each parameter on the honey flow outcome. Each year, between 300 and 600 colonies were observed. The results of this study show that the population factors which explained the weight of harvested honey are first, the amount of capped brood and, to a lesser extent, the number of bees. A maximum Varroa load of 3 Varroa mites per 100 bees at the beginning of the honey flow was identified as the threshold beyond which colonies performance are weakened. These long-term observations have provided a general background that beekeepers use to improve colony preparation for this honey flow.

**Abstract:**

In response to the concerns of beekeepers on the decline of honey bee populations on lavender honey flow in the lavender fields of southeast France and the consequent decrease of honey production, our long-term survey (2009–2021) monitored the total weight gain collected by these colonies. This study shows the variations in the total weight gain according to regions, years, populations structure (bee number and quantity of capped brood) and Varroa load. Among these factors, years and regions support one third of the variations over this 13-year survey. At the beginning of the honey flow, capped brood is more important than the number of bees, whereas Varroa load severely limits the performance of the colonies. A threshold of 3 mites/100 bees seems to reflect the upper limit of the Varroa load below which the total weight gain is not affected. This survey provides useful information for the beekeepers to better prepare the colonies for this honey flow and allows them to compare their results obtained with our general description of the total weight gains by year.

## 1. Introduction

The worldwide observation of the decline of honey bee colonies is primarily reported by the death of colonies during the winter [1]. Beekeepers cope with this by reinforcing the replacement of dead colonies for maintaining the level of their stocks of colonies at the beginning of spring. However, the role that these decline factors play in seasonal honey production is poorly documented. During the years 2003–2006, the beekeepers who set up apiaries on lavender in the southeast of France reported severe declines of hive populations and in the final harvest of lavender honey [2]. These observations echoed the observations from around the world reporting declines in bee populations. Sudden depopulations (whose biological causes inside the colonies are diverse [3,4,5]) and environmental or climate changes (whose causes and consequences are beyond the reach of beekeepers’ actions [6,7]) tend to generate catastrophic discourse on colony performance. Yet, there is a gap between the decline of honey bees reported in the literature and the variations in colony productivity observed by beekeepers. In this context, the collection of beekeeper data should allow for rational analyses leading to more accurate discussion. However, statistics applied to participatory data reveal severe accuracy problems, great difficulties in interpretation and considerable methodological complexity [8].

In the particular case of the southeastern region of France, where lavender and lavandin are important honey crops, beekeepers wanted to avoid the hazards of participatory investigational methods based on the personal beekeeper records and, therefore, asked for a longitudinal, observational study on the honey flow on lavender based on objective measurements of the weight gain of colonies during the honey flow. An observational study requires at least three characteristics:-Observations extending over a sufficient number of individuals to be able to estimate the intra-individual variability with sufficient precision so as not to degrade the statistical results;-Ample time periods to assimilate intra-annual variability;-Sufficiently extended areas to absorb spatial variations in the environmental conditions of the object observed. In the case of professional apiaries, a fourth characteristic is required: the longitudinal survey must be aimed at a large group of beekeepers that allows for a diversity of beekeeping practices to be considered.

In this observational configuration, the researchers construct the protocol, collect the data and carry out the analysis and synthesis. Beekeepers provide production apiaries with healthy colonies. The choice of the geographic positions of apiaries in the honey flow area and the number of colonies monitored per apiary are the main qualities of the observational study that guarantee good conditions for statistical analysis and interpretations.

The main aim of this study is to evaluate the scope of effectiveness of beekeepers to manage their colonies in order to obtain the best harvests under the spatial and temporal variation of resources.

## 2. Materials and Methods

The lavender honey flow observational study was initiated in 2009 and operated continuously until 2021, in the southeast of France.

For 13 years, 268 apiaries were surveyed. There are three distinct regions of lavender production: Drôme Provençale, Plateau de Valensole and Montagne de Lure et Plateau d’Albion (Figure 1). Altogether, 4726 colonies have been monitored.

### 2.1. Data Collection

At the beginning of the lavender honey flow, starting between the 8th of June and the 1st of July depending on the year and the region, the number of worker capped brood cells (excluding drone brood) and the number of bees (only after 2012) were estimated with the ColEval method [9]. At the same time, phoretic mite numbers were determined using soap washes of 300 bee samples collected in all colonies [10]. Samples of bees were collected from brood areas containing emerging adult honey bee workers [11,12]. Varroas nested it the brood cells were not accounted for. The brutal decrease of the number of capped worker brood occurred at the end of the lavender honey flow and the resulting release of Varroa mites from the brood at that time had little effect on the weight gain of the colonies.

Additionally, hive bodies and supers were weighed separately at the beginning and at the end of honey flow, before the final harvest. Supplementary supers were considered as void when added, with a weight of 6.5 kg. Colony performance is described by the total weight gain, i.e., the weight difference between the end and the beginning of the honey flow.

### 2.2. Statistical Analysis

To frame the study of the precursors of colony productivity, we proposed a general model based on foraging ecology [13,14] that describes the relationship between colony performance (=total weight gain over the entire period of honey flow) and three main classes of factors: the resource, the forager population and the health status of the colony, written as follows (for hive *j* in apiary *i*):Performance*_ij_* ~ *f* (Resource*_i_* + Population*_ij_* + Health Status*_ij_* + error)

The factors in this model can be divided by those outside the influence of beekeepers (e.g., weather conditions, landscape complexity, wild for cultivated floral diversity) and those within the control of the beekeepers (e.g., size of the colony population, mite control).

All statistical analyses and figures were generated in R [15]. The general model proposed above is adapted to the survey of the lavender honey flow as follows (model 1):(1)$$\begin{array}{c} y_{k}=\gamma_{0}+\gamma_{1}\;u_{1j1\left[k\right]}+\gamma_{2}u_{2j2\left[k\right]}+\gamma_{3}u_{3j3\left[k\right]}+\beta_{1}\;x_{1k}+\beta_{2}\;x_{2k}+\beta_{3}\;x_{3k}\\ +\nu_{1j1\left[k\right]}+\nu_{2j2\left[k\right]}+\varepsilon \end{array}$$
where:*y_k_*: the total weight gain (*TWG*) for the hive *k*;$\gamma_{0}$: the overall intercept;$\gamma_{1}\;u_{1j1\left[k\right]}$: the intercept due to the predictor *u*_1_ of group *j*_1_ (=year);$\gamma_{2}\;u_{2j2\left[k\right]}$: the intercept due to the predictor *u*_2_ of group *j*_2_ (=region);$\gamma_{3}\;u_{3j3\left[k\right]}$: the intercept due to the predictor *u*_3_ of group *j*_3_ (=apiary);$\beta_{1}\;x_{1k}+\beta_{2}\;x_{2k}+\beta_{3}\;x_{3k}$: the direct effect of variable *x*_1_ (=brood), *x*_2_ (number of honey bees) and *x*_3_ (number of mites/100 bees);$\nu_{1j1\left[k\right]}+\nu_{2j2\left[k\right]}+\gamma_{3}u_{3j3\left[k\right]}$: random variation of groups *j*_1_, *j*_2_ and *j*_3_;ε: the residual standard error.

The total weight gain and the population variables (brood, number of bees and number of mites/100 bees) were measured at the colony level. We do not have any resource measurements or indices at the colony level or at the apiary level. We avoid this difficulty by considering apiary as a grouping factor of hives. All of the hives of one given apiary, for one year and one region, are under the same environmental conditions; moreover, each apiary is not only under the specific conditions of its particular location but also under the influence of beekeeper management strategies and practices (i.e., genetics, preparation of hives, migrations that occur before lavender honey flow, etc.). In our study, colony population structure at the time of the settlement of apiary is considered as the starting point, not taking into account for the specific characteristics: queen age, spring nourishment, reinforcement of population, etc.).

The apiary as a grouping factor in the model supports a larger part of the random variability (which includes the annual and regional variations) in the model and it leads to a more statistically powerful estimation of the direct effect of the variables measured at the hive level.

The model is then transformed as follows (model 2):(2)$$y_{k}=\gamma_{0}+\beta_{1}\;x_{1k}+\beta_{2}\;x_{2k}+\beta_{3}\;x_{3k}+\nu_{1j\left[k\right]}+\varepsilon_{k}$$
where:*y_k_*: the total weight gain for the hive *k*;$\gamma_{0}$: the overall intercept;$\gamma_{1}\;u_{1j1\left[k\right]}$: the intercept due to the predictor *u*_1_ of group *j*_1_ (=apiary);$\beta_{1}\;x_{1k}+\beta_{2}\;x_{2k}+\beta_{3}\;x_{3k}$: the direct effect of variable *x*_1_ (=brood), *x*_2_ (number of honey bees) and *x_3_* (number of mites/100 bees);$\nu_{1j1\left[k\right]}$: random variation of group *j*_1_;ε: the residual standard error.

In the model used for evaluating population and health effects on the total weight gain, a linear mixed model (LMM) with Identity as the link function was used. All apiaries were considered as independent by year and region and then apiary was chosen as a random effect applied to each variable. The R function lmer {package: lme4} was used.

The preliminary studies of the effect of the grouping factors (year, region and apiary) on the total weight gain were made with a linear model (R function lm {package: stats}.

$R^{2}$ coefficients for the linear mixed model were estimated using the function r.squaredGLMM {package MuMin} [16].

## 3. Results

### 3.1. Annual Variations in Colony Productivity

The average performance as measured by total weight gain across our 13-year survey over the three regions is 26.36 ± 13.53 kg/year.

The range of annual variations of the total weight gain is clearly depicted in Figure 2. A pairwise Student test showed four classes of performance:

2009, 2017 and 2019 as low-performance years (*a*);

2010, 2015, 2018 and 2020 as high-performance years (*b* and *bc*);

2011, 2013 and 2021 as medium-high-performance years (*c* and *cd*);

and 2012, 2014 and 2016 as medium-low-performance years (*de* and *e*).

Additionally, the absence of a noticeable trend in the total weight gain over 13 years is consistent with regard to the absence of trends in the number of honey bees or the quantity of capped brood (see Figure 1 and Figure 2).

### 3.2. Regional Variations in Colony Productivity

Productivity among the three regions appear to be significantly different across all years of the survey (Figure 3).

On average, the total weight gain is about 3.87 ± 1.2 kg higher in the Drôme region than in the Lure Albion region and higher by 6.7 ± 1.2 kg in the Valensole region than in the Drôme region.

The overall differences between the three regions mask some specific annual particularities such as where the total weight gain in the Valensole is lower than that in the Drôme (in 2012) and even lower than that in the Lure Albion (in 2016).

### 3.3. Effect of Capped Brood Quantity at the Beginning of Honey Flow

In order to simplify the presentation of descriptive results, capped brood quantity, number of bees and the number of mites per 100 bees have been grouped into classes. Classes were fixed according to a simple description convenient for beekeepers rather than by calculating equal-measure density in each class.

The positive effect of the quantity of capped brood at the beginning of honey flow on lavender is clearly shown in Figure 4.

The step of 2500 cells is about the number of cells that covers two-thirds of the full frame of capped brood for a Dadant body frame; it represents almost full coverage of a Langstroth body frame [9]. The classes are organized into four groups but show a linear increase in total weight gain with the quantity of brood at the beginning of honey flow.

### 3.4. Effects of the Number of Bees at the Beginning of Honey Flow

Similarly, grouping the number of bees by classes of 5000 bees shows the effect of the size of the honey bee population on total weight gain (Figure 5). The increasing step of 5000 bees is approximately the number of bees that could be found on two faces of a frame in Dadant hives and on three frame faces in Langstroth hives.

### 3.5. Effect of Varroa Load at the Beginning of Honey Flow

The Varroa load at the beginning of the honey flow is grouped into classes of unequal load according to the information used by beekeepers to evaluate the critical steps of Varroa population increase along the season [17]. The first four classes where mites/100 bees are less than three can be considered as the Varroa loads that do not have a diminishing effect on the total weight gain (Figure 6).

### 3.6. Combined Relative Effects of the Different Colony Parameters on Colony Productivity

The effect of grouping factors describes the context in which the total weight gain of the colonies of one apiary is obtained. Neither “year” nor “region nor “apiary” are independent. Variance analysis is then not possible between these groups. Nevertheless, the simple linear regression between the total weight gain and each of these three grouping factors gives information about their respective contribution to the variability of total weight gain. In the following simple linear model, y is the response variable (total weight gain), x is the group factor (year, region or apiary) and *ε* is the residual. Additionally, a new variable is defined as year by region (=AnReg) where each of the individuals is independent.
(3)$$x\;\sim \;\gamma_{0}+\gamma_{1}\;x+\varepsilon$$

In Table 1, the $R^{2}$ for the three group factors is calculated by: $$R^{2}=\frac{\sigma^{2}\;\left(\hat{y}\right)}{\sigma^{2}\;\left(y\right)}$$
with $R^{2}$ giving the proportion of the variability of y explained by the group factor x.

From Table 1, the interaction of factors year and region explains the maximum proportion of the response variable y (32.68%).

The difference between the proportion of the response variable y explained by “AnReg” and by “apiary”, apart from the local environmental conditions, could be regarded as the effect of the beekeeper practices, which include all of the techniques for the preparation of colonies in anticipation of the lavender honey flow but also the experiences of the colonies in an apiary before being brought to the lavender honey flow, such as the previous honey flows. On average, the proportion of the variability due to beekeeper practices could be around 25.12% (*R*^2^_apiary_ − *R*^2^_anreg_).

For estimating the effect of the population factors and the Varroa load on colony performance, model 2 is used (Table 2). As the honey bee populations were evaluated starting in 2013, the model is run only on this part of the data set. As a result of the model 2, the conditional $R^{2}$ of the model is 0.7281; the marginal $R^{2}$ of the direct effects only is 0.2227; and, in contrast, the $R^{2}$ of the group effect “apiary” is 0.5054.

This means, in practical terms, that:-each additional 10,000 capped brood cells are responsible for 6.91 ± 0.307 kg of total weight gain;-each additional 10,000 honey bees are responsible for 4.076 ± 0.649 kg of total weight gain;-when the number of mites/100 bees increases by 1, the total weight gain of one colony is decreased, on average, by 0.7192 ± 0.065 kg.

The values of the standard errors are noticeably small, which means that the coefficients describe the benefit that beekeepers could have in taking into account these factors as important predictors of the colony productivity during the lavender honey flow.

If the model takes into account only the capped brood and the number of mites/100 bees and then is applied over the 13 years, the results are very similar:-capped brood: 8.837 × 10^−4^ ± 2.676 × 10^−5^;-number of mites/100 bees: −6.404 × 10^−1^ ± 5.481 × 10^−2^.

## 4. Discussion

This analysis has described the hierarchy of the different factors influencing total weight gain during the honey flow on lavender. The main argument of this hierarchy is that each of the factors contributes to the variability of the response variable.

First of all, the better model (model 2) has a global $R_{global}^{2}$ of 0.728 that leaves unexplained more than 25% of the variability of total weight gain. The rest of the variability is mainly due to the characteristics of apiary location and beekeeping practices ($R_{LMMapiary}^{2}$ = 0.5054). From $R_{LMapiary}^{2}$ calculated in the linear model used for the calculation of the interaction effect of year and region, it can be inferred that the $R_{Year*Region}^{2}$ due to environmental conditions accounts for around three-fifths of $R_{LMMapiary}^{2}$ (i.e., 30%) and that the $R_{practices}^{2}$ is about 20%. Finally, the direct factors (capped brood, number of bees and Varroa load) under the control of beekeepers at the time of lavender honey flow explained 22.27% of the variability of the total weight gain.

In these results, apart from the influence of environmental factors, the total weight gain of one hive during the complete honey flow for one given year results from a complex trade-off between, at a minimum, the amount of collected nectar, the intensity of overnight drying of nectar in the hive and the honey consumption by the bee population of the colony. The variations across years or regions reflect mainly the availability of the resources, but not only, since the environmental conditions govern the concentration of nectar, the foraging ability of worker bees and the food needs of the colony. Moreover, the total weight gain does not exactly depict the amount of honey available for the beekeepers because a part of the total weight is stored in the hive body. Finally, the reader must keep in mind that these results are strictly related to the lavender honey flow in the south of France and cannot be generalized to the global activity of honey bees, neither for the whole apicultural season nor for other geographical areas. Despite more brood and bees and less Varroa being obvious factors that influence the activity of colonies and then their productivity, the values of the coefficient characteristic of these factors are really governed by the temporal regime of the resource variations. Lavender honey flow is remarkably long (more than 3 weeks), which explains that the amount of worker capped brood is essential in providing young bees which can be foragers during the second part of the honey flow. In the case of short honey flows (e.g., sunflower or black locust tree), the number of bees at the beginning of honey flow is a better precursor than capped brood [18].

The residual error of the model with direct effects and group factor (model 2) (e.g., the part of the variability otherwise unexplained) is greater than 25%, which means that, despite the fact that group factor carries a large part of the variability linked to the environmental conditions, unknown individual colony behaviors are important. The evaluation of the direct effects of population and health factors over such a long period could minimize the specific effect, in each year and at each location, of one of these factors. Performing a more detailed yearly analysis focusing on the effects of these factors could contribute to a better understanding of the interaction between the two categories of factors (population and Varroa load), especially in explaining certain cases in which a heavy Varroa load did not lead to a decrease of total weight gain, perhaps due to the strong colony dynamics that overcame pressure by Varroa.

The successive locations where the migratory apiary are settled in spring and summer aim to seek out continuous resources. Equally in the sedentary apiary, resource uncertainty can lead the colonies to a lack of food supply and then to a weakening of the population [19], which means that the colonies that arrive at the lavender flow are in various stages of brood or honey bee dynamics regardless of the number of these components.

The direct positive effect of the amount of capped brood and of workers bees on honey production were clearly demonstrated. The weight gains appeared lower than those described in previous studies [20] but this is mainly due to the high variability captured by large number of colonies monitored and the long duration of the survey [21]. The additional weight gain by 10,000 capped brood cells or by 10,000 honey bees (7 kg and 4 kg respectively) must be compared to the global average of the total weight gain over the whole survey (26.4 kg) because the values of 7 kg and 4 kg are the expected mean of the probability distribution of the GLMM coefficients. Colony size at the beginning of the honey flow are essentials factors of the honey production [22] provided that the health, and especially the Varroa load, of the colony is well controlled by the beekeeper. The reason for the effect of the quantity of capped brood could be related to the duration of the lavender honey flow (between 25 and 32 days). The newly laid brood at the time of the beginning of honey flow will provide a sufficient number of foragers to ensure the second part of the honey flow.

Across the 13 years of this observational study, it is not possible to observe a general increasing or decreasing trend in the amount of honey stored in the hives. This is one of the most important results to be pointed out, as the general opinion on honey bee activity is expected to be deceasing as a consequence of the decline of bee populations and bee activities. Nevertheless, if a trend does exist, due to the decline of honey bee activity or the global changes which act upon resource availability, the variability of the total weight gain over years is large enough to mask it. A longer survey could probably better answer whether or not a trend exists.

Several attempts have been made along the 13-year survey to determine which agro-environmental factors (development of lavender plants, modification of the land uses around the apiaries, resource availability before lavender honey flow, etc.) could be linked to the average variations of total weight gain between years or regions (unpublished data). However, no statistically relevant relationships which outweigh the effects of starting number of honey bees, capped brood or Varroa load, have been found. The variation of the resources and foraging conditions, depicted by the variation of the average total weight gain per year, probably has a more complex determination. Additionally, the phenological stage of the lavender plantation, and also the nectar availability at the beginning of the honey flow, depend on the climatic conditions that govern the growth of lavender plants from the autumn season before the observed honey flow. Altogether, the intensity of honey flow seems to be unpredictable. That is why, knowing the large variation of foraging success for honey bees by years and regions, it is recommended to allocate the apiaries in the different regions.

Noticeably, these results are specifically linked to lavender honey flow in the south of France. Any rapid generalization will be at risk of missing the complexity of each honey flow. A similar survey [18], of a lesser extent of time, that is currently being conducted on sunflower honey flow in the southwest of France (7 years of measures; 200 to 240 hives surveyed each year) gives dissimilar results, especially in pointing out a more important role of number of bees than of capped brood.

In the first period (2009–2013), this long survey has provided the beekeepers with useful information about the colony population structure favoring the total weight gain during the lavender honey flow. The beekeepers have progressively increased the quantity of capped brood of the hives prepared for lavender honey flow. The quantity of capped brood has been stabilized since 2013 [21]. In the second period, until recently, the beekeepers requested the continuation of the survey of the lavender honey flow for it provided a spatio-temporal context of the colony performance, which is helpful for them in analyzing their own results in comparison to the results from the survey.

Despite the fact that, in the overall average, a hierarchy is observed within the three regions of lavender production in the southeast of France, annual variations may reveal exceptions to this average tendency. For this reason, beekeepers are encouraged not to concentrate all of their apiaries in the same region. As a consequence of the long duration of this survey, the beekeepers have expressed hope that it could be used as a tool for understanding climatic evolution in the weather conditions during the honey flow.

## Figures and Tables

**Figure 1 insects-13-00472-f001:**
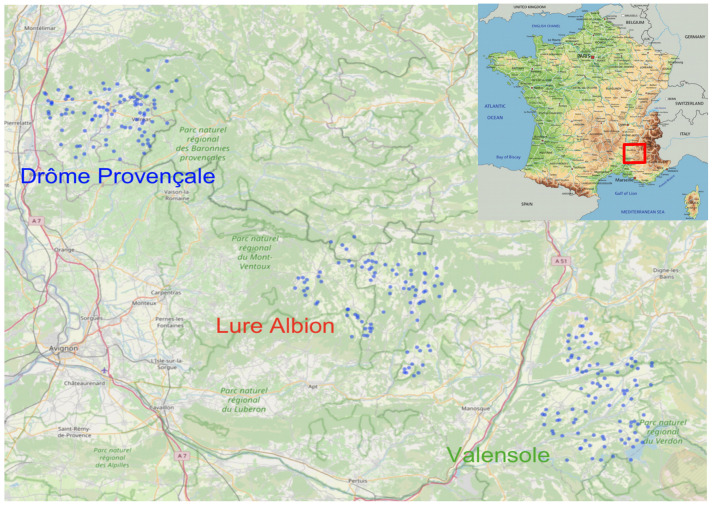
Map of the location of the apiaries surveyed during the 13 years. Each blue dot represents one apiary.

**Figure 2 insects-13-00472-f002:**
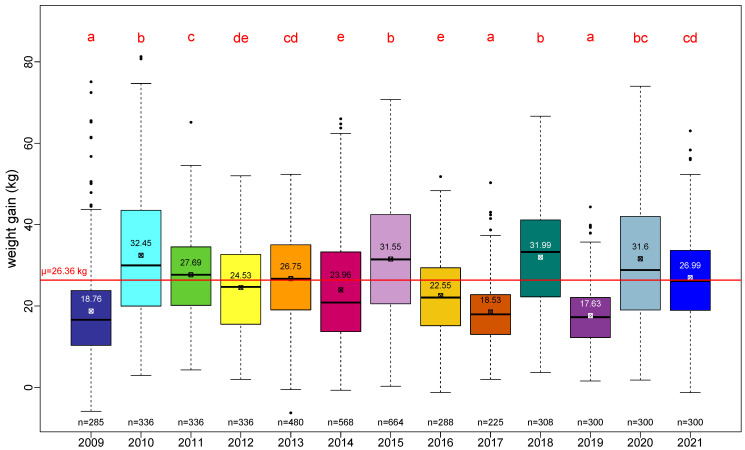
Total weight gain per year across the three regions. Red letters result from a pairwise Student test (*p*-value < 0.05). The red line represents the overall average of total weight gain at the colony level. Number in each box plot is the average total weight gain for that year. Number “n=” below each box plot is the number of colonies surveyed for that year.

**Figure 3 insects-13-00472-f003:**
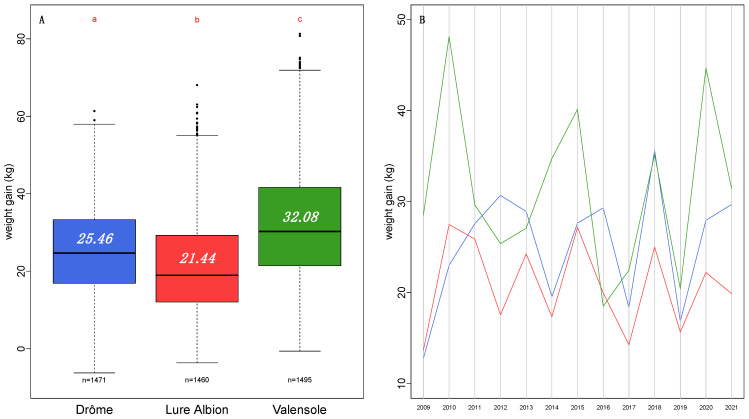
(**A**) Total weight gain per region across the 13-year survey. Red letters result from a pairwise Student test (*p*-value < 0.05). Number in each box plot is the average total weight gain for that region. Number “n=” below each box plot is the number of hives surveyed for that region. (**B**) Average annual variation in the average total weight gain for the three regions (blue = Drôme; red = Lure Albion; green = Valensole).

**Figure 4 insects-13-00472-f004:**
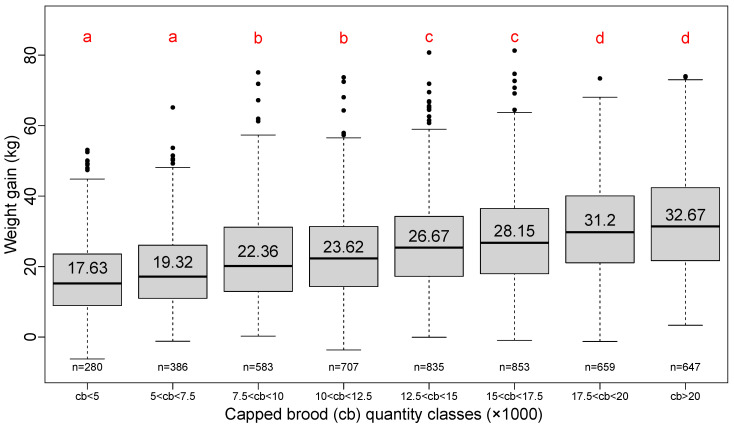
Boxplot series of eight classes of capped brood quantity. Step of increase: 2500 cells of capped brood. Red letters result from a pairwise Student test (*p*-value < 0.05). Number in each box plot is the average total weight gain for that class. Number “n=” below each box plot is the number of hives surveyed for that class. Annual variations are shown in Appendix A: Figure A1.

**Figure 5 insects-13-00472-f005:**
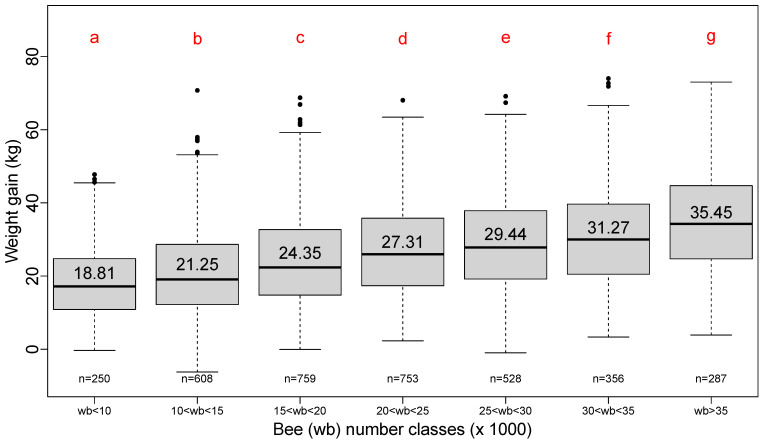
Boxplot series of seven classes of honey bee numbers. Step of increase: 5000 bees. Red letters result from a pairwise Student test (*p*-value < 0.05). Number in each box plot is the average total weight gain for that class. Number “n=” below each box plot is the number of hives surveyed for that class. Annual variations are shown in Appendix A: Figure A2.

**Figure 6 insects-13-00472-f006:**
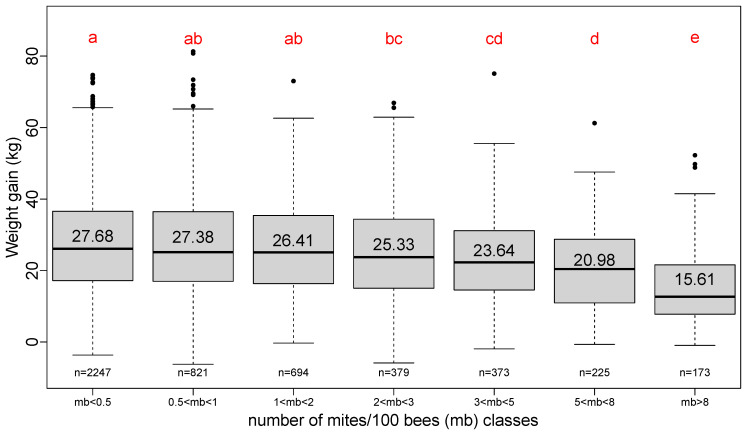
Boxplot series of seven classes of number of mites/100 bees. Unequal step of increase. Red letters result from a pairwise Student test (*p*-value < 0.05). Number in each box plot is the average total weight gain for that class. Number “n=” below each box plot is the number of hives surveyed for that class.

**Table 1 insects-13-00472-t001:** Estimation of the proportion of the response variable y by each of the group factors (year, region, apiary, AnReg).

Groups	*p*-Value	$\sigma^{2}\;\left(y\right)$	$\sigma^{2}\;\left(\hat{y}\right)$	$R^{2}$
Year	$<2.2\times 10^{-16}$	183.10	23.39	0.1255
Region	$<2.2\times 10^{-16}$	183.10	19.23	0.1047
Apiary	$<2.2\times 10^{-16}$	183.10	110.44	0.578
AnReg	$<2.2\times 10^{-16}$	183.10	60.84	0.3268

**Table 2 insects-13-00472-t002:** Coefficients and standard errors for the fixed effects of model 2.

Fixed Effects	Coefficient	Standard Errors
Worker capped brood	$6.919\times 10^{-4}$	$3.065\times 10^{-5}$
Honey bee number	$4.076\times 10^{-4}$	$1.956x\times 10^{-5}$
Mites per 100 bees	$-7.192\times 10^{-1}$	$6.495\times 10^{-2}$

## Data Availability

All data are available on the site “Apimodel”: http://w3.avignon.inra.fr/lavandes/biosp/observatoires.html (last update: 3 May 2022).

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
