# Peer review of "More Worker Capped Brood and Honey Bees with Less Varroa Load Are Simple Precursors of Colony Productivity at Beekeepers’ Disposal: An Extensive Longitudinal Survey"

_insects, 2022, doi:10.3390/insects13050472_

Round 1
Reviewer 1 Report
Overall, a well written paper covering what appears to be an extensive and robust dataset which examines effects of various factors on lavender honey production in the Southeast of France. An informative, yet concise, read. Good care was taken in accounting for standard sources of variability in sampling, which can inform those planning to examining other similar systems. As the dataset is based on one region and foraging source, the authors have taken care not to overstate the scope and impact of their model, nor their findings. In order to be accepted for publication, I suggest the authors address the following minor issues:
Line 39: Clarification on what is meant by reconstruction of stocks (line 39)
Line 42: Clarify what is meant by “final harvest”. I assume honey, just make clearer.
Line 46: Parenthesis close, rather than open.
Line 63: Discussion of intra-annual variability. Perhaps sampling frequency is a better descriptor than “time periods”?
Line 101-102: What is meant by the supplementary supers being considered void? Does this mean that any honey collected in them would be standardised to 6.5kg, even if filled/empty?
Model 1 and Model 2: Was drone presence accounted for? ColEval method only examines worker numbers, yet it’s not clear if “brood” in model relates to only workers, or worker and drone. Could define brood more clearly in methods.
Whilst apiary is a good indicator of overall environmental factors and management practices, it doesn’t necessarily account for those hives that have had new queens introduced or the method of doing so (buying mated vs. own hive production). Make clearer if this was accounted for.
Whilst no. of phoretic Varroa is measured, there’s no indication that those within capped brood (~80% of total popn) are accounted for. Make clear why decision was made to only account for phoretic mites.
Consider colouring different region apiaries in figure 1 to match colours in figure 3.
Table title above table
All figures: remove graph titles and integrate into figure legends.
All figures: move sample number “n” down, so as to not obstruct error bars and outlier point. Similarly for grouping letters, move up to set above outliers.
Figure 6: x-axis labels could be made clearer.
Slight repetition in the discussion with regards to limitations of the analysis. Consider combining.
Author contributions: resources listed as “X.X”
Author Response
Responses to the reviewer 1.
Line 39 : change done
Line 42 : change done
Lne 46 : change done
Line 63 : change done
Line 101,102 : yes ; sentence changed to be more precise
Drone brood :Precisions on drone brood given (line 120)
Beekeeper practices : details given (line 178)
Varroas nested in brood : explanations given (line 125)
Figure 1 : changed
Table title place above
Titles of graphs : removed and integrated in legends (line 326)
Figures changed so as to not obstruct error bars and outlier points
Reviewer 2 Report
Summary: This manuscript describes the results of a 13 year study of honey bee hive condition and productivity in southern France surrounding the lavender nectar flow. Hives that had more capped brood, more adult workers, and fewer Varroa mites produced more honey on average. This data can be used by beekeepers to help boost their hive productivity by monitoring and managing their hives for stronger, more productive colonies.
This is a really great data set!
Major points:
- Overall: The manuscript needs general review for clarity and confusing word choice throughout. Also the words “of” and “to” are frequently used in inappropriate places.
- Formatting: I don’t know if this happened during the submission process or if it was from the original document, but the formatting is weird. The entire manuscript is split into sort of mini-paragraphs making it choppy to read.
- Title: Please make the title more clear.
- The simple summary is too simple. Please provide broader context and more specific results. Clarify throughout.
- The abstract needs more broader context and definition of the problem.
Line comments:
Simple summary
Line 14: What do you mean by “how much leeway”?
Line 15: Word choice “population components”. Maybe “factors”, or “variables”?
Line 15-16: Word choice “weight”. In this context, it’s confusing to use the term “weight” since here it does not refer to hive weight. Perhaps change to “importance of each parameter to predict honey flow”.
Line 17-18: Flip this sentence so start with “Honey flow was explained by…”
Line 18: Explain what you mean by “maximum threshold”. Threshold for what?
Abstract
Line 21: Need to add a broader context before the first line here. Also, is there a documented or perceived decline in honey bees in lavender?
Line 23: Change “collected of” to “collected by”
Line 23: Change “variations of” to “variations in”
Lime 30: Change “colonies to” to “colonies for”
Introduction
Overall: The introduction is broken into short mini-paragraphs. Needs to be made more cohesive with improved flow.
Line 42: Final harvest of what? Honey? Clarify.
Line 57: Clarify “hazards of participatory investigational methods” with some examples.
Line 66: What do you mean by “observatory”?
Line 72: Clarify what you mean by “choice of the distribution of apiaries”.
Materials and methods
Line 84: What do you mean by “localization”? I was not able to view the appendix.
Fig. 1: Add an inset of France showing where in the country the study region is located.
Line 108: Simplify “To frame the study of the precursors of colony productivity…”
Line 117: Can you give examples of the “resource” variables? Does this mean weather, landscape, etc.? Maybe start this paragraph with something like “The factors in this model can be divided by those outside the influence of the beekeepers (e.g., weather), and those within the control of the beekeeper (e.g., mite control).”
Line 136: The font is weird for “random variation of groups”
Line 158/Model 2: I don’t think all of the variables are listed within the model.
Results
Line 176: Change “performance of total weight gain” to “performance as measured by total weight gain”
Line 177: kg/year?
Figures: The figure titles have weird font and spacing
Line 198: Change “The three regions” to “Productivity among the three regions”
Line 236: Write out/define TWG
Line 241: I think you mean 5000 bees, not 500 cells of brood
Line 258: Change “Region nor” Apiary” to “Region” nor “Apiary”
Line 279: Clarify this sentence. Maybe flip so “Interaction of Year and Region” is first.
Line 286-287: Where does “25.12%” come from? I don’t see that in the table.
Line 292-299: This might be better in a table format.
Line 301-307: These results seem really important, especially for the beekeepers. Can you highlight this better? Right now it’s kind of hidden.
Discussion
Overall: What’s up with the formatting? Why so many hard returns throughout?
Line 315: Change “factors of total” to “factors influencing total”
Line 316: Change “for the honey flow” to “during the honey flow”
Line 338: “These results are strictly related…cannot be generalized”… I think you can talk about limitations of the study here but I would think that these results would definitely be applicable in other systems. Why do you think these results are not generalizable?
Line 343: Need a better topic sentence here. Also change “>” to “greater than”
Line 353: What do you mean by “different stages”?
Line 374-379: Paragraph starting “Across the 13 years” I don’t understand what this paragraph is saying.
Line 382: “agro-environmental factors” like what? Give some examples.
Line 384: Word choice “overtake”. “Outweigh”?
Line 393: What do you mean by “organize the distribution”?
Line 405: Change “level of the quantity” to “quantity”
Line 506: “Appendix”?
Author Response
Responses to the reviewer 2.
Title changed
Simple Abstract
Line 14 : sentence changed
Line 15 : word changed as suggested
Lines 15-16 : sentence changed
Lines 17-18 : sentence rewritten
Line 18 : sentence changed
Abstract
Line 23 : context added ; references of honey bees decline are given in the introduction.
Line 23 and 30 : changed done
Introduction
Line 23 : Text format changed
Line 42 : changed
Line 57: changes done
Line 66 : changes done
Line 72 : change done
Materials and methods
Line 84 : change done
Fig 1 : change done
Line 108 : sentence split
Line 117 : precision given( line 148-150 new MS)
Line 136 : fixed
Line 158 : precision given (line182 new MS)
Results
Line 176 : change done
Line 177 : change done
Figures titles removed
Line 198 : change done
Line 236 : done
Line 241 : yes ; done
Line 258 : change done
Line 279 : change done
Line 286-286 : explanation given
Line 292-299 : results given with table format as requested
Line 301-307 : Details given to highlight the results
Discussion
Text format revised
Line 315 : change done
Line 316 : change done
Line 338 : explanation given.
Line 343 : explanation given and change done
Line 353 : sentence changed and explanation given
Line 374-379 : more explanation are given to make the paragraph clearer
Line : 382 : exalples given
Line 384 : change done
Line 393 : change done
Line 405 : change done